# Elucidation of Early Evolution of HIV-1 Group M in the Congo Basin Using Computational Methods

**DOI:** 10.3390/genes12040517

**Published:** 2021-04-02

**Authors:** Marcel Tongo, Darren P. Martin, Jeffrey R. Dorfman

**Affiliations:** 1Center for Research on Emerging and Re-Emerging Diseases (CREMER), Institute of Medical Research and Study of Medicinal Plants (IMPM), Yaoundé, Cameroon; 2Division of Computational Biology, Department of Integrative Biomedical Sciences and Institute of Infectious Disease and Molecular Medicine, Faculty of Health Sciences, University of Cape Town, Cape Town 7925, South Africa; darrenpatrickmartin@gmail.com; 3Division of Medical Virology, School of Pathology, Faculty of Health Sciences, Stellenbosch University, Cape Town 7505, South Africa; jeffreydorfman@gmail.com

**Keywords:** HIV-1 group M (HIV-1M), Congo Basin, phylogenetic, sequences, evolution

## Abstract

The Congo Basin region is believed to be the site of the cross-species transmission event that yielded HIV-1 group M (HIV-1M). It is thus likely that the virus has been present and evolving in the region since that cross-species transmission. As HIV-1M was only discovered in the early 1980s, our directly observed record of the epidemic is largely limited to the past four decades. Nevertheless, by exploiting the genetic relatedness of contemporary HIV-1M sequences, phylogenetic methods provide a powerful framework for investigating simultaneously the evolutionary and epidemiologic history of the virus. Such an approach has been taken to find that the currently classified HIV-1 M subtypes and Circulating Recombinant Forms (CRFs) do not give a complete view of HIV-1 diversity. In addition, the currently identified major HIV-1M subtypes were likely genetically predisposed to becoming a major component of the present epidemic, even before the events that resulted in the global epidemic. Further efforts have identified statistically significant hot- and cold-spots of HIV-1M subtypes sequence inheritance in genomic regions of recombinant forms. In this review we provide ours and others recent findings on the emergence and spread of HIV-1M variants in the region, which have provided insights into the early evolution of this virus.

## 1. Introduction

The simian ancestors of HIV-1 group M (HIV-1M) have been transmitted to humans from Chimpanzees (SIVcpz) in south eastern Cameroon in the Congo Basin (CB) near the beginning of last century [1,2]. It is very plausible that during the very earliest stages of the epidemic, the HIV-1M progenitor adapted to, and was transmitted locally between, multiple Cameroonian individuals. This localized epidemic would have been followed by a period of expansion within an increasingly connected African population when one or more HIV-1M-infected individuals travelled or immigrated in the 1920s to the city of Kinshasa in the Democratic Republic of Congo (DRC) [3,4,5].

By the 1950s, repeated human-to-human transmissions related to several human and socio-demographic factors such as high-risk behaviour and iatrogenic interventions within this city had allowed the virus to establish and evolve in human and probably yielded a diverse HIV-1M population [6,7,8]. Among the HIV-1M genomes circulating at that time, four of them led to the HIV-1M subtype A, B, C and D lineages that today are the dominant drivers of the global AIDS epidemic [9,10,11]. Different HIV-1M lineages started to colonise the world from the CB [12,13], for example, entering the Americas in 1966 [14] and East Africa around 1970 [8]. However, all the variants arising in the CB were not part of this initial migratory wave and remained confined to the region. Two elements are considered key players in the dissemination of HIV-1M. First, the sociological and historical factors. These include population migrations, economic development and urbanization and inter and intra state wars [15]. The second factor implies that the demographic composition of HIV-1M populations in different parts of the world may simply reflect the timing and frequency of HIV-1M dispersal or founder events [6,8]. Beside these unpredictable stochastic processes, it is also possible that there were some viral features and host factors that may have influenced the rise of some variants in, and their spread out of the CB. Specifically, it is possible that, the precursors of today’s HIV-1M main subtypes, under positive Darwinian selection acting between its first transmission to humans (in ~1900) and the onset of the global epidemic (in ~1950), experienced a succession of genetic changes that facilitated their dissemination from the CB region. The best way to test such a hypothesis would be to directly compare the biological properties of the HIV-1M sequences that are ancestral to the globally circulating HIV-1M variants (such as subtypes A, B and C) with those that are ancestral to variants that have remained largely restricted to west Africa (such as subtypes J, H, and K). As HIV was only discovered in the beginning of the 1980′s, the prospect to achieve this goal is extremely limited. Nevertheless, the recent development of computational sequence analysis tools and the accumulation of contemporary sequences from the CB region have greatly improved our understanding of viral genetic events that may have made this epidemic possible. Specifically, these computational tools have (1) provided evidence-based interpretations of the extent of HIV-1M diversity in the early phases of the epidemic [16]; (2) indicated how HIV-1M genomes may have been assembled through recombination prior to the discovery of HIV-1M [13]; and (3) indicated which regions of the HIV genome will probably remain least prone to recombination in the future. Recombination is also a major mechanism for maintaining genetic diversity of other viruses; a process which increases the rate of adaptive evolution in response to changing environments and vaccine-induced immune responses (Reviewed by [17]). From the perspective of designing globally relevant biological interventions, it is particularly important to fully understand the biological underpinnings of HIV-1M diversity: how it arose, spread and impacted the extents and durations of individual HIV-1 sub-epidemics. This review provides an overview of the key findings pertaining to the early evolution of HIV-1 M in the CB region.

## 2. HIV-1M Lineages Diverging Early after the Cross-Species Transmission Event

As HIV-1M was only discovered in the early 1980s, our directly observed record of the epidemic is largely limited to these past four decades. However, the contemporary HIV-1M variants that were generated so far, have encoded within their genomes a significant amount of information about the evolutionary and epidemiologic history of the virus. Using these modern sequences, phylogenetic analysis have suggested that there is growing evidence both that presently unidentified HIV-1M lineages were likely epidemiologically relevant during the early stages of the HIV-1M epidemic and that recombinant fragments of some of these lineages might still be circulating within the CB region. The first evidence for this was provided by Carr et al. [18]. They phylogenetically analysed two sets of sequences generated from HIV-infected individuals residing (1) in remote villages and (2) in Yaoundé, the cosmopolitan capital city of Cameroon. They found that compared to that of the cosmopolitan city, recombinant lineages from the remote villages were comprised of complex mosaics of sequences that had apparently been derived from divergent HIV-1M lineages most closely related to rare subtypes like F2, H and K [18]. Since such recombination would have required frequent coinfections of individuals with divergent HIV-1M variants, Carr et al. [18] questioned how fragments of genome sequences derived from subtypes that were believed to circulate at very low frequencies could have made their way into so many contemporary recombinant HIV-1M genomes. They concluded that some of these mosaic viruses, rather than simply being recombinants of viruses in contemporary HIV-1M lineages, may represent the descendants of rare pre-epidemic HIV-1M lineages, i.e., in a sense these divergent HIV-1M variants were “evolutionary relics” [18].

Similar findings were later published by Villabona Arenas et al. (2017). They performed two surveys on antiretroviral drug resistance in the DRC and identified several sequences that fall basal to the subtype C sub-tree [19]. After characterization of near full length genome of some of these lineages, they identified tree mosaic recombinants including two novel CRFs (92_C2U and CRF93_cpx) with segments with unknown origin and others that were located at the base of subtype C, A and CRF02_AG sub-trees [19].

Furthermore, in one of the most extensive studies performed to date throughout Angola (another country in the CB), Bartolo et al. (2009) characterized 159 HIV-1M sequences collected from HIV-infected individuals and found that 8% of these sequences could not be classified within any known HIV-1M lineages [20]. Additionally, they identified two highly divergent sub clusters in the subtype A radiation [20]. It should be noted that just because a new sequence is most closely related to sequences belonging to a particular known subtype or CRF does not necessarily mean that this new sequence should represent a virus that is genuinely highly divergent. It is also possible that this sequence originates from a recombinant virus that branches at the outskirts of known clades [21]. One should therefore be aware that recombination can bias reconstruction of a phylogenetic tree and that special care should be taken when analysing data set from rapid evolving organism like HIV [22].

We have adopted an original approach to better identify and characterize the highly divergent HIV-1M sequences found in the CB [23]. We first designed a method to select representative sequences of all the subtypes and CRFs together with other published divergent HIV-1M sequences. Instead of just a random selection based on country of origin or date of collection, we ensured that the representative sequences were specifically selected to include the broadest diversity of sequences previously identified as belonging to known HIV-1M subtypes and CRFs. To achieve this breadth of sequence diversity for each subtype/CRF, we first retrieve all available sequences for each individual subtype and CRF from the Los Alamos National Library (LANL); we then construct a maximum likelihood (ML) trees for each subtype/CRF and selected one sequence from each of the most basal lineages from the root of these subtypes and CRFs depending on the desired number of representatives (Figure 1) [23]. We next performed a fully exploratory screen for recombination that did not assume the existence of known parental sequence lineages (as is commonly the case when people characterize HIV-1M recombination) [16]. Specifically, we tested all sequences for evidence of both intra- and inter-clade recombination, regardless of whether they had formerly been classified as pure subtypes, CRFs or URFs. Furthermore, the tool that we used for this recombination screen, RDP4 [24], could be used to automatically decompose recombinant sequences into their different constituent parts. As a consequence, we were able to generate a mostly recombination-free sequence dataset, i.e., a multiple sequence alignment from which most or all recombinationally derived sequence fragments had been removed (Figure 2).

All our ML downstream trees were generated using RAxML version 8 [25]. Although RAxML version 8 is limited to the use of GTR-based nucleotide substitution models, it has been specifically designed to accurately infer phylogenies from alignments containing large amounts of missing data [26]: a factor ideally suiting to the analysis of an alignment generated from sequences sets that have gaps where recombinant fragments were separated. Finally, we defined more accurately divergent sequences as either (i) those residing on isolated branches outside of sub-trees containing previously defined HIV-1 subtype or CRF lineages, or (ii) those clustering with low degrees of associated bootstrap support within sub-trees containing previously defined HIV-1 subtype or CRF lineages (Figure 3) [23].

We then generated a ML tree with RAxML version 8 and showed that some parental lineages of certain CRFs contained more than 7000 bp that are not classifiable within the currently established HIV-1M subtypes, suggesting that they are predominantly descended from what were/are major previously unidentified HIV-1M lineages that were likely epidemiologically relevant during the early stages of the HIV-1M epidemic [16].

This suggested that large pools of undiscovered HIV-1M genetic diversity likely existed, and potentially still exist, throughout equatorial West Africa either as entire genomes or as fragmentary sequences within recombinant lineages.

What is clear is that these divergent lineages appear to have not undergone the same explosive global spread as other HIV-1M subtypes: perhaps because they were less transmissible or perhaps because they were not in the right place at the right time. Concerted efforts to characterize more of these divergent lineages will add additional branches to the base of the HIV-1M phylogenetic tree that will substantially increase the power with which phylogenetics-based analytical methods can infer the selection patterns, demographic processes and geographical range dynamics occurring before the HIV-1M pandemic began. Crucially these same computational techniques could in principle, enable the accurate inference of actual ancestral genome sequences of the major known HIV-1M lineages, thus allowing these ancestral genomes to be resurrected using chemical synthesis and characterized in the laboratory to directly determine their biological characteristics.

## 3. Diversification and Dissemination of HIV-1M Subtypes

Several studies have attempted to reconstruct the early dynamics of the different HIV-1M subtypes and other CRFs that are circulating in the CB [15,27,28,29,30]. Using similar Bayesian statistical approaches, these studies have inferred that different lineages emerged at different locations in the region between the 1950s and 1970s and then spread out of the region. For example, while subtype G likely originated from Cameroon in 1953 [27], the most recent common ancestors (MRCAs) of all the CRF01_AE and CRF02_AG viruses were, respectively located in the Central African Republic in 1970 [30] and the DRC [31] in 1973.

In one of the most extensive studies of this kind, Faria et al. (2019) investigated the evolutionary dynamics of HIV-1M subtypes A1, C, D, F1, H and J using a set of data comprising about 350 *pol* sequences. These samples were collected from HIV infected individuals from several places in the DRC. While they traced the origin of HIV-1M A1 and D in the city of Kinshasa between 1950 and 1960, respectively, they also found that subtype C originated in a southern location of the DRC in the 1950s before moving to other places of the Central and East Africa; subtypes H and J, respectively originated in the seaport city of Matadi much more later [15].

In an earlier study, Delatorre and Bello (2016) explored the time-scales over which some HIV-1M complex circulating recombinant forms arose within the CB region, including CRF- 09_cpx, 11_cpx, 13_cpx and 45_cpx. Surprisingly, they reported that these mosaic lineages have been circulating in the region for a period comparable to the early circulation of the now prevalent HIV-1M lineages [28]. In this regard, they have traced the time of the MRCAs (tMRCAs) of CRFs 09_cpx, 11_cpx and 13_cpx to 1966, 1957 and 1965, respectively. Although it cannot be discounted that the estimated dates when the MRCAs of the analysed recombinants existed might actually represent the tMRCAs of their parental viruses [32] (such that the actual recombination events that gave rise to these clades could have happened later), this study provides more evidence that highly divergent HIV-1M lineages were able to establish epidemiologically successful strains during the early stages of the epidemic in the CB.

Furthermore, using a Bayesian coalescent-based method, Delatorre and Bello (2016) also investigated the dissemination dynamics of the CRF11_cpx clade and found that the epidemic growth of the clade was similar to that of other HIV-1M lineages circulating in the CB [28]. This suggests that CRF11_cpx, and perhaps other complex HIV-1M recombinant variants within the CB region, have likely been adapting to a genetically consistent human population for far longer than HIV variants found in other, more cosmopolitan parts of the world.

We have also investigated the diversification and dissemination of HIV-1M subtype A [13]. For this analysis we used a comprehensive set of both published near full-length subtype A sequences, and subtype A derived genome fragment sequences found within circulating CRFs and URFs. This was done by manually removing recombinant sequences from our alignment and splitting them into their constituent recombinationally derived fragments based on previously inferred breakpoint locations. Individual subtype A-derived fragments larger than 1000 nt in length were then re-added to the initial alignment, with gap characters being added to the 3′ and 5′ ends of the fragments to ensure that they remained correctly aligned with the rest of the dataset. This enabled us to infer that the MRCA of all presently sampled subtype A viruses likely existed sometime around 1942 in the CB region, shortly before the divergence of the main subtype-A lineage into at least four sub-lineages (A1 to A4).

While sub-groups one and two contain isolates previously described as sub-subtype A1 and A2, respectively, sub-group three contains all isolates previously identified as belonging to sub-subtype A4 and the final sub-group, A5, contains A-attributed genome segments from CRF26_AU isolates. It was apparent that the best sampled of these sub-lineages, A1, has been a major contributor genomic sequence to the genomes of several CRFs. Further, CRFs 02, 37 and 45 each likely have multiple different subtype-A1 parental viruses suggesting they were likely sequentially assembled during multiple different mixed infections that included divergent A1 lineage virus We could also detect the presence of subtype A in West Africa and Cuba as early as 1974, in Eastern Europe in 1977, Western Europe and Central Asia around 1988 [13].

The accuracy of the time-scale and location of the MRCA of different lineages largely depends on the sets of sequences that are analysed; it is common to find discrepancies in the locations and dates of particular ancestral sequences across studies. This is well illustrated with estimates of the date when the subtype G MRCA existed. We analysed the set of available subtype G full-length *env* sequences that were publicly available at the time of the analysis, including subtype G-derived and G-like *env* sequences, supplemented by full length *env* sequences that we derived from Cameroonian blood samples and a very few partial sequences from Angola that were publicly available [27]. We determined a tMRCA of the subtype G clade between 1939 and 1963 [24] that was earlier than previously published tMRCA for subtype G [33,34]. Several explanations for this discrepancy are possible; however, the most parsimonious explanation is that we included more sequences related to older lineage(s) than the previous studies or included sequences related to a broader range of lineages. These explanations are not mutually exclusive. It was not possible to resolve the role of Angola or DRC because we had so few Angolan or DRC-derived *env* sequences; however, one Angolan sequence did appear basal to the large cluster of Spanish/Portuguese sequences in the Bayesian analysis.

The subtype G *pol* sequence analysis of Delatorre et al. [34] also suggested that the origin of this subtype may have been in Angola/DRC/Republic of Congo (which they analysed as a single region) with movement into Cameroon—a movement that could not have been resolved with our *env* sequences because of how few subtype G *env* sequences are available from Angola, the DRC and the Republic of Congo. Similarly, a lack of subtype G *env* sequences sampled from Nigeria also means that analysis of this genome region could also not be used to detect the movement of subtype G into, and expansion within, Nigeria.

With respect to the early history of subtype G *env*, CRF06_cpx *env* sequences were embedded within subtype G, while CRF25_cpx *env* sequences were basal to subtype G, suggesting that the CRF25_cpx *env* was likely derived from an HIV-1M lineage related to the MRCA of subtype G that has remained undiscovered and may be extinct. Therefore, although these subtype G sequence analyses have filled some gaps in our knowledge of the early events in the spread of HIV-1M, the lack of whole genome sequence data is still seriously undermining our ability to definitively date the origins and track the early diversification of this important lineage.

## 4. HIV-1M Recombination and Adaptation

Recombination is a key characteristic of HIV-1M, shaping its evolution, diversity and adaptation. According to the LANL [35], there are today more than a hundred known CRFs and uncountable known URFs. Hemelaar J, et al. (2020) updated the current contribution of different subtypes and recombinants to the global HIV-1M epidemic between 1990 and 2015; they found that CRFs accounted for about 23% to all HIV-1M infections worldwide during that period [36]. This proportion is even higher in some regions such as East and Southeast Asia, and West and Central Africa where CRFs are the most prevalent lineages [36,37].

Few studies have dealt with the challenge of determining the recombination histories of the major HIV-1M lineages (“clades”) themselves. We tried to fill the gap, by studying the history of subtype A sequences and subtype A-derived sequence fragments from recombinant genomes before and during its dissemination throughout much of the world [13]. Using a Bayesian statistical approach implemented in the software BEAST v1.8.4 [38], we presented data suggesting that globally, two periods of high rates of recombination might have shaped the diversity of subtype A. The first period was between the mid-1960s and mid-1970s and involved mostly viruses with complex mosaic structures including CRFs_ 04_, 06_, 09_, 13_, 18_, 37_ and 45_cpx. The second period occurred after the onset of the global pandemic during the 1970s and includes CRFs_ 50_A1D, 03_AB and 32_06A1. These findings are consistent with the hypothesis that, at the onset of the global HIV epidemic, subtype A, and more particularly sub-subtype A1, viruses circulating in the CB were genetically predisposed to successfully found HIV-1 epidemics in other parts of the world [13], possibly more so than many other subtypes.

Whereas some other subtypes such as B and C may have been, like subtype A, predisposed to founding new epidemics, subtypes like G, may have been less predisposed to this. For example, despite subtype G arriving in Europe at approximately the same time as the now dominant subtype B subtype B [27], it has remained largely restricted to Spain and Portugal and represents only a minor part of the European HIV-1M epidemic.

The extensive contributions that subtype A sequences have made to the known HIV-M recombinants is not mirrored in subtype C and subtype B sequences. Additionally, it remains unknown why subtype A- derived sequences appear so frequently in recombinants that likely arose before the discovery of HIV-1.

It is important to note that the recombination events that we are able to resolve have been filtered by natural selection such that recombinants comprising sequences from other subtypes could arise as frequently as those containing subtype A derived sequences but have simply not survived selection. Additionally, the extent to which different potential parental lineages circulated within the same human groups may not have been spatially and/or temporally consistent between the different subtypes. It may never be possible to differentiate between these possibilities without a library of sequences that date to the period during which subtype A sequences contributed so extensively to the recombinants that we see today.

Nevertheless, the large numbers of recombinant forms containing fragments of subtype A derived sequences have prompted speculation as to whether the subtype A derived sequences within these recombinants have imbued with selective advantages over non-recombinants.

Several studies have sought to determine the relative fitness of different HIV-1M subtypes and HIV-1M recombinants. For example, Koulinska et al. (2006) presented data on a Tanzanian study cohort, where they compared cases and controls of mothers who transmitted the virus through breastfeeding to their infants. Using a multivariate analysis, they found that inter-subtype recombinants between subtypes A, C, or D were apparently more transmissible through breast-feeding than were pure subtype A, C, and D viruses [39]. If these recombinants are indeed fitter than their parental viruses, it is remains unclear why they have not expanded relative to pure subtype A, C and D viruses in southern Tanzania. One possibility is that the selective advantage(s) of the recombinants is restricted to mother to child transmission.

Another study performed by Kouri et al. [40] on a Cuban cohort of HIV-1-infected individuals showed that, CRF19_cpx, a complex recombinant between subtypes D, A and G originating from the CB, were among the several factors that were found to be statistically related to the rapid progressor condition. However, it is not immediately obvious how being associated with rapid progression to AIDS might contribute to increased fitness.

## 5. HIV-1M Recombination and Evolution

Additional studies have tried to map recombination breakpoint distributions across the HIV-1M genome by analysing large numbers of recombinant sequences [41,42,43]. They provided an agreement that these distributions appear non-random with notable recombination breakpoint hot-spots occurring near the 5′ and 3′ ends of env and cold-spots occurring within the gp120 encoding region [41]. Furthermore, Golden et al. [44] have determined that the distribution of recombination across the HIV-1M is strongly influenced by purifying selection acting against the survival of recombinants that have disruptions of amino acid interactions within folded proteins and/or disruptions of nucleotide interactions within the folded genome secondary structures. HIV genomes may have even evolved over the long-term to ensure that biochemically predisposed recombination hot-spots correspond to genome sites at which recombination-induced protein and/or genomic nucleic acid folding will be minimally disrupted.

Another study determined the relative frequencies with which nucleotide sequences in different parts of recombinant genomes have been derived from parental genomes belonging to the different HIV-1M subtypes [45]. In this study, we analysed the distribution within 283 CRFs and URFs, of genome fragments derived from HIV-1M Subtypes A, B, C, D, F, and G and CRF01_AE. Counts were made along the alignment of the number of times individual nucleotides along the genomes of viruses in the different subtypes were inherited by recombinant genomes; a permutation test was performed to determine whether certain nucleotides were more frequently inherited by recombinant genomes from parents belonging to a particular subtype than could be accounted for by chance under random fragment exchanges (Figure 4). We identified statistically significant hot-spots of subtype A sequence inheritance in genomic regions encoding portions of *gag* and *nef*; this might indicate that the subtype A portion of those genes is in some way superior to those of viruses in the other HIV-1 subtypes. Viruses belonging to the other analysed HIV-1M subtypes and CRF01_AE also contributed certain genome fragments more frequently during recombination than other fragments [45]. Furthermore, we have searched in the literature studies that have tried to quantify differences in the relative functionality of individual genes from viruses belonging to different HIV-1M subtype and found that this non-randomness in the frequencies with which different subtypes have contributed specific genome regions to known HIV-1M recombinants is consistent with selection strongly impacting the survival of inter-subtype recombinants.

It is entirely plausible that this non-randomness in fragment exchanges amongst different HIV-1M subtypes is a consequence of Darwinian selection processes that have favoured the survival of recombinants containing particular fragments of sequences from particular subtypes. In line with this, Arenas et al. (2016) have demonstrated the relevance of quasispecies heterogeneity in the evolution of HIV-1 in cell culture. To this end, they have analysed the contribution of mutation and recombination in HIV-1 viral fitness in a cell culture. They found an increased fitness of the viral population after several passage in the cell culture, which correlated with an increase in quasispecies heterogeneity with the appearance of fixed mutations and recombination that favour more adapted variants [46]. It is also likely that that the observed patterns were strongly influenced by the demographics of viral populations circulating at the time when these recombination events occurred. It cannot be discounted, therefore, that the observed recombination patterns may simply reflective a predominance of particular subtypes within mixed infections that occurred during the 1960s and 70s rather than particular genome fragments of these subtypes possessing some innate fitness-enhancing qualities.

## 6. Conclusions

Genomic nucleotide sequence analyses have been a useful tool to infer HIV-1M past demography and historical events during the early phases of the HIV-1M epidemic. They are key to understanding how the virus evolves, spreads and diversifies. In this review, we have summarised data aiming to retrace the movement and evolutionary dynamics that were at play during the earliest stages of the HIV-1M pandemic in the CB region. We found that many genomes originally inferred to be from recently recombinant lineages contain fragments from evolutionarily early lineages that likely predate the HIV-1 migratory wave that triggered the global epidemic in the 1950s. The restriction of these divergent lineages to the CB region suggests that they were less infectious and/or simply not present at the time and place of that initial migratory wave. Some subtypes such as A seem to have been involved in many of the early major dispersal events and have provided genome fragments that feature prominently within many of the known HIV-1M recombinants. In contrast, other subtypes such as G, J and K have made a far more modest contribution to the global pandemic. Although it remains to be definitively determined whether such differences are attributable to some subtypes being innately more predisposed to founding new epidemics than others, increased sampling of whole genome sequences within the CB region may eventually reveal enough about the early stages of the pandemic to address this question.

## Figures and Tables

**Figure 1 genes-12-00517-f001:**
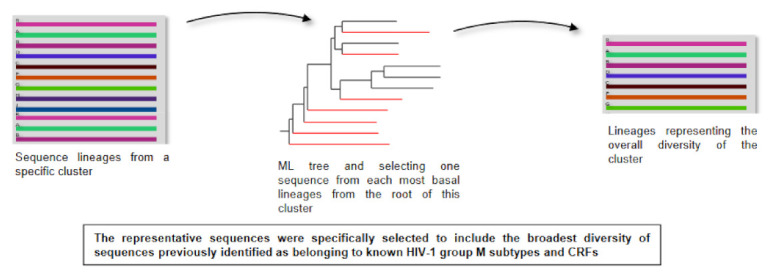
Example of how a representative selection of sequences from each subtype and CRF was achieved. First, a maximum likelihood tree from all 12 sequences of this subtype was constructed, and then, a selection of one sequence from each of the up to seven most basal lineages from the root was made as a representative sample of the overall known diversity of this particular subtype. Copyright © Oxford University Press, Evol Med Pub Health 2015(1):254–265 (2015), used under Creative Commons CC-BY license.

**Figure 2 genes-12-00517-f002:**
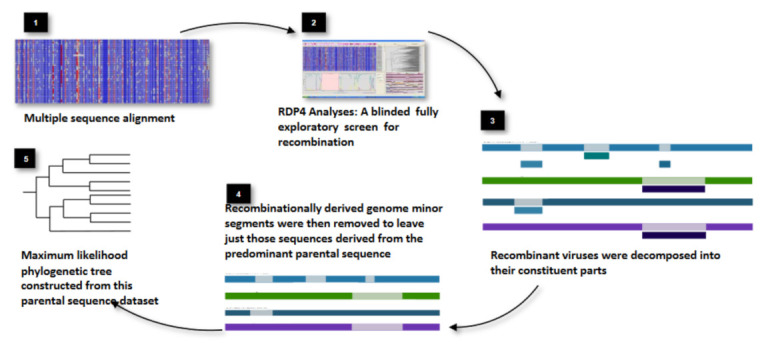
Schematic representation of how a recombination-free tree is constructed using RDP4. Copyright © American Society for Microbiology, Journal of Virology 90:2221–2229 (2016), used with permission.

**Figure 3 genes-12-00517-f003:**
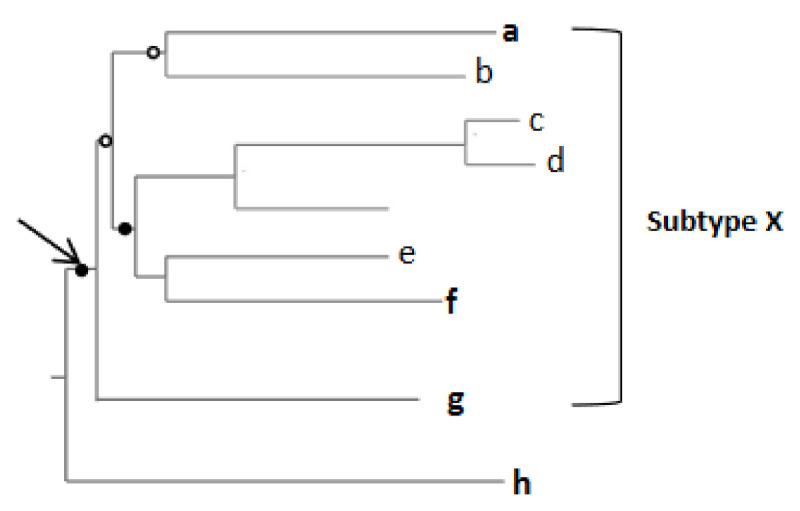
Schematic illustration of highly divergent sequences. In this example, a, f, g and h (in bold) represent new sequences from a subtype, X; the solid circle represents a bootstrap value of >70% while an open circle represents a value of <70%. The arrow shows the basal node of subtype X. In this example, ‘a’ clusters with ‘b’ but with <70% bootstrap support; in addition, they form a subtree which also contains ‘c’, ‘d’, ‘e’ and ‘f’ with a <70% bootstrap support; therefore, ‘a’ represents an example of a divergent sequence likely clustering near the base of subtype X (it is also possible though that it clusters nearer the crown). Since the bootstrap support for the branch containing the subtree with ‘c’, ‘d’, ‘e’ and ‘f’ is >70%, ‘d’ is considered to be embedded within the subtype X clade. ‘g’ is located at the base of this subtype. Sequence ‘h’ is a divergent lineage branching outside of subtype X and is likely the extant descendent of a lineage that diverged prior to the diversification of the subtype X MRCA. Copyright © Oxford University Press, Evol Med Pub Health 2015(1):254–265 (2015), used under Creative Commons CC-BY license.

**Figure 4 genes-12-00517-f004:**
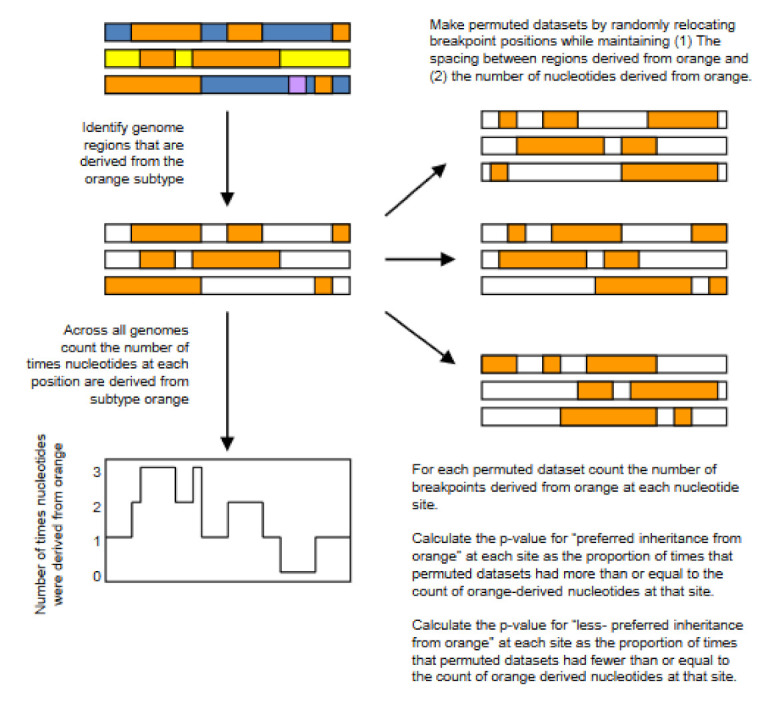
The permutation test used to determine whether certain nucleotides within recombinant HIV genomes have been inherited from parental viruses from particular subtypes more or less frequently than can be accounted for by chance. The nucleotides derived from different subtypes are indicated by different colours. In this case we are interested in nucleotides derived from the orange subtype. Copyright © Oxford University Press, Virus Evolution 4(1):vey015 (2018), used under Creative Commons CC-BY NC license.

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
