# Peer review of "Elucidation of Early Evolution of HIV-1 Group M in the Congo Basin Using Computational Methods"

_genes, 2021, doi:10.3390/genes12040517_

Round 1

Reviewer 1 Report

HIV-1-M subtype has been the primary source of HIV infection worldwide. The authors have provided a comprehensive review of the previous studies and their own studies in investigating the biological basis of HIV-1M diversity, such as how it arises, spreads, and impacts the extent and duration of individual HIV-1 sub-epidemics. They also reviewed and provided a broad overview of essential findings in HIV-1 M subtypes' ancestral evolution in the Cuban Basin. This review will help search for previously rare HIV recombinant subtypes that may play an essential role in high divergence and adaptation in the Viral genome in different regions of the Congo basin and Worldwide. Also, they performed their own study to identify and characterize these highly divergent or mosaic (containing different subtypes including the rare/ previously unknown) recombinant sequences. This study is important in understanding the HIV-1-M virus genomic diversity and how they evolve from their progenitors.

I did not find any significant revision to be made. However, I recommend a few minor revision -
1) There is no line number. It is difficult for a reviewer to point out anything in the text if they want it to be revised by the author.
2) Please provide a version number for the tools that the authors used. For example, the author mentions that they used RaxML for generating the ML tree. Please provide the version number of RaxML. Similarly, please provide the version number of RDP4.
3) Authors mentioned in their figure legend that "the solid circle represents a bootstrap value of >70% while an open circle represents a value of <70%." Instead of representing the bootstrap score by the solid and open circle, I would recommend using the actual bootstrap score for each node." It will improve the understanding of the level of bootstrap support for each clade in the tree and will make the study look more reliable. 

Author Response

Reviewer 1:

Comments and Suggestions for Authors

HIV-1-M subtype has been the primary source of HIV infection worldwide. The authors have provided a comprehensive review of the previous studies and their own studies in investigating the biological basis of HIV-1M diversity, such as how it arises, spreads, and impacts the extent and duration of individual HIV-1 sub-epidemics. They also reviewed and provided a broad overview of essential findings in HIV-1 M subtypes' ancestral evolution in the Cuban Basin. This review will help search for previously rare HIV recombinant subtypes that may play an essential role in high divergence and adaptation in the viral genome in different regions of the Congo basin and Worldwide. Also, they performed their own study to identify and characterize these highly divergent or mosaic (containing different subtypes including the rare/ previously unknown) recombinant sequences. This study is important in understanding the HIV-1-M virus genomic diversity and how they evolve from their progenitors.

I did not find any significant revision to be made. However, I recommend a few minor revision -

  • There is no line number. It is difficult for a reviewer to point out anything in the text if they want it to be revised by the author.

This is an appropriate point and we fully agree on this comment. Unfortunately, this does not depend on us, but rather to the editorial secretariat. We submitted our manuscript as per journal guidelines which do not require such a formatting. 

  • Please provide a version number for the tools that the authors used. For example, the author mentions that they used RaxML for generating the ML tree. Please provide the version number of RaxML. Similarly, please provide the version number of RDP4.

This was done for RAxML (version 8). The “4” in the RDP4 already mentions the version.

  • Authors mentioned in their figure legend that "the solid circle represents a bootstrap value of >70% while an open circle represents a value of <70%." Instead of representing the bootstrap score by the solid and open circle, I would recommend using the actual bootstrap score for each node." It will improve the understanding of the level of bootstrap support for each clade in the tree and will make the study look more reliable.

The tree in the figure is just an illustrative example on how to best identify a “divergent” lineage in a phylogenetic tree and was not constructed using existing sequences; therefore we cannot have numbers as bootstrap value in the different nodes.

Reviewer 2 Report

Genes-1116311; Tongo et al.

This is a nicely written review that reveals the complexities and inherent limitations associated with this field of investigation (complexity of recombination patterns; limited access to sequences from viral variants that were in circulation early on during the pandemic; complexity of regional and worldwide patterns of dispersion; necessary reliance on intricate bioinformatics approaches). References include recently published work in the field (e.g. Faria et al., PLoS Pathog 15: e1007976, 2019; Hemelaar et al., J Virol 95: e01580-20, 2020). One excellent scholarly account of the early history of the HIV-1 pandemic can be found in Pepin J, The Origins of AIDS, Cambridge University Press, 2011 (ISBN: 9781139005234; DOI: https://doi.org/10.1017/CBO9781139005234).

It would have been nice if the authors had provided original figures instead of lifting them from their previous papers.  

Author Response

Reviewer 2:

Comments and Suggestions for Authors

Genes-1116311; Tongo et al.

This is a nicely written review that reveals the complexities and inherent limitations associated with this field of investigation (complexity of recombination patterns; limited access to sequences from viral variants that were in circulation early on during the pandemic; complexity of regional and worldwide patterns of dispersion; necessary reliance on intricate bioinformatics approaches). References include recently published work in the field (e.g. Faria et al., PLoS Pathog 15: e1007976, 2019; Hemelaar et al., J Virol 95: e01580-20, 2020). One excellent scholarly account of the early history of the HIV-1 pandemic can be found in Pepin J, The Origins of AIDS, Cambridge University Press, 2011 (ISBN: 9781139005234; DOI: https://doi.org/10.1017/CBO9781139005234).

Many thanks. The last reference was also added

It would have been nice if the authors had provided original figures instead of lifting them from their previous papers.

 We actually provided original figures.